# A Quaternion-Based Piecewise 3D Modeling Method for Indoor Path Networks

**Hongdeng Jian [1], Xiangtao Fan [1,2], Jian Liu [1,\*], Qingwen Jin [1,3] and Xujie Kang [1,3]**

[1]  Key Laboratory of Digital Earth Science, Institute of Remote Sensing and Digital Earth,
    Chinese Academy of Sciences, Beijing 100094, China; jianhd@radi.ac.cn (H.J.); fanxt@radi.ac.cn (X.F.);
    jinqw@radi.ac.cn (Q.J.); kangxj@radi.ac.cn (X.K.)
[2]  Key Laboratory for Earth Observation of Hainan Province, Sanya Institute of Remote Sensing,
    Hainan 572029, China
[3]  University of Chinese Academy of Sciences, Beijing 100049, China
[\*] Correspondence: liujian@radi.ac.cn; Tel.: +86-10-82178079

**Abstract:** Generating 3D path models (with textures) from indoor paths is a good way to improve the visualization performance of 3D indoor path analysis. In this paper, a quaternion-based piecewise 3D modeling method is proposed to automatically generate highly recognizable 3D models for indoor path networks. To create such models, indoor paths are classified into four types of basic elements: corridor, stairs, elevator and node, which contain six kinds of edges and seven kinds of nodes. A quaternion-based method is devised to calculate the coordinates of the designed elements, and a piecewise 3D modeling method is implemented to create the entire 3D indoor path models in a 3D GIS scene. The numerical comparison of 3D scene primitives in different visualization modes indicates that the proposed method can generate detailed and irredundant models for indoor path networks. The result of 3D path analysis shows that indoor path models can improve the visualization performance of a 3D indoor path network by displaying paths with different shapes, textures and colors and that the models can maintain a high rendering efficiency (above 50 frames per second) in a 3D GIS scene containing more than 50,000 polygons and triangles.

**Keywords:** indoor path network; 3D modeling; quaternion; path analysis

## 1. Introduction

3D path network analysis for indoor space provides strong decision support for users in searching for optimal routes in applications such as emergency services, disaster management, security, transportation and visitor guiding [1–3]. Implementing indoor path analysis in 3D GIS will help users understand the results and make decisions intuitively by visualizing the virtual city environment, buildings, indoor scenes, paths, nodes, and optimal routes [4–7]. The buildings and indoor scenes can be visualized in various ways, such as detailed solid models, wireframes, boxes shaded with textures, and even translucent meshes. However, 3D paths are usually visualized simply as 3D points and lines [2,7,8], which are able to describe the paths' shapes but make it difficult to identify the types. It would be better to generate 3D models (with textures) from the 3D points and lines, so that we can distinguish the various types and complex components of indoor paths.

Generating 3D models from lines automatically is a common task in graphical modeling and computer-aided design, e.g., in generation of pipelines [9,10] and machine parts [11–13]. There are many ways of calculating the coordinates of model vertices in 3D model generation, such as Euler angle methods and quaternion methods. Compared to the Euler angle calculation, quaternions are much more flexible and elegant in coordinate transformation, 3D modeling and animation [14–16].

In this paper, we propose a quaternion-based method of calculating the coordinates of indoor path elements (indoor corridors, stairs, elevators and nodes) according to the complex indoor path structure and advantages of quaternion operations. Afterwards, a piecewise 3D modeling procedure was implemented to automatically create the complete 3D models from indoor paths. Based on these methods, 3D indoor path models were generated from an indoor path network and applied to path analysis. Five 3D scene primitives (vertices, lines, polygons, triangles and quadrangles) in different visualization modes were compared, and frame rates of four stages (without paths, paths visualized as lines, paths visualized as models and path analysis with 3D models) were recorded to analyze the efficiency of the proposed method. The results indicate that this quaternion-based piecewise 3D modeling method could improve the visualization performance of a 3D indoor path network by displaying paths with different shapes, textures and colors and that the method could maintain a high rendering efficiency in a 3D GIS scene. Therefore, the method proposed in this paper will have promising prospects for path analysis related applications in Digital Earth and Smart Cities [4,17–20].

This paper is organized as follows. The next section introduces several related studies, including the visualization of 3D indoor path analysis and 3D model generation methods using lines. Section 3 describes data curation and methodology, including classification and development of an indoor path network, design of basic elements, the quaternion-based 3D modeling method and the piecewise 3D modeling procedure. Section 4 presents the results of 3D indoor path modeling and path analysis, and analyzes the method's efficiency. The paper concludes with the discussion in Section 5.

## 2. Related Work

### 2.1. Visualization of 3D Indoor Path Analysis

A 3D GIS integrated model using CityGML and Oracle Spatial was developed to visualize a building model and implement 3D network analysis [4]. The building model was stored in the CityGML format and could be visualized in four different types: WireFrame, HiddenLine, Shaded, and Shaded with Texture. The indoor path network model was stored in Oracle Spatial and represented by linear networks (lines and points) in the 3D scene, and the shortest path analysis result was represented by thick black lines. An extended semantic model based on CityGML was proposed to improve the indoor navigation results by dividing the concept Space into three concepts: room, corridor and stair [6], which would clearly identify the types of spaces along the shortest path but fail to convey the types of paths because the path network was not semantically segmented, and the paths were represented by lines. The IFC (Industry Foundation Classes) standard was introduced to path plans for 3D indoor spaces because of its capability to restore both geometric information and rich semantic information of building components [7]. The geometric and semantic information of indoor space was extracted from the IFC file and transformed into 3D building models, and the shortest path was visualized as dotted or solid lines. The combined use of IndoorGML and Land Administration Domain Model (LADM) was explored to define the accessibility of indoor spaces based on the access rights for spaces [3]. The space of a building was divided into corridors, stairs, lifts, toilets, etc., and visualized with different colors, but the path network was still represented by lines.

The visualization of 3D indoor path analysis in the existing studies is mainly concerned with the (semantic) division and representation of indoor space, which indeed will help understand the process and results of path analysis; however, the paths are usually visualized as 3D lines that cannot clearly indicate the type of path. The semantic idea can also be introduced to classify the indoor path network and subsequently facilitate the visualization of indoor path analysis by generating 3D models (with textures) from 3D points and lines.

### 2.2. 3D Model Generation Methods from Lines

Approaches to generating 3D models from lines vary, and include Euler angle and quaternion methods. Euler angle methods usually involve many complex calculations of trigonometric

functions [10,13,21], which makes the process of angle calculation difficult to understand and rather cumbersome to implement, especially in a 3D scene. Additionally, different rotational sequences can result in different 3D models based on Euler angles, so a 3D model based on Euler angles is not uniquely defined and sometimes inconvenient [14]. The concept of quaternion, first described by Irish mathematician William Rowan Hamilton in 1843 [22], is a complex digital system consisting of three complex numbers and one real number:

$$Q[x, y, z, w] = x \cdot \vec{i} + y \cdot \vec{j} + z \cdot \vec{k} + w \tag{1}$$

$$\vec{i}^2 = \vec{j}^2 = \vec{k}^2 = -1 \tag{2}$$

where $x$, $y$, $z$ and $w$ are real numbers, and $\vec{i}$, $\vec{j}$ and $\vec{k}$ are imaginary unit vectors.

A quaternion can express the rotation of any vector axis, and it is more efficient and flexible than the method of Euler rotation and matrix rotation. It can also avoid the problem of gimbal lock. A 3D model based on quaternions is uniquely defined because it does not depend on the rotational sequence. Thus, quaternions have been widely used in modeling, rendering, animation [15,23,24] and object control, such as that of spacecraft [14,25,26] and robots [27,28]. Generating 3D path models from indoor path networks using quaternion-based methods would be convenient and efficient.

## 3. Materials, Concepts and Methods

### 3.1. Classification and Development of an Indoor Path Network

Building a 3D topological network with nodes and edges is a fundamental task of 3D path analysis. To create detailed and vivid 3D models for the path network, the nodes and edges should be classified into different types, according to their shapes and functions. Considering that pedestrians can only move along specific paths, such as corridors and stairs, in an indoor scene, indoor paths are classified as three types in this paper: Indoor corridors, stairs and elevators. These edges and the nodes among them interconnect to form a complete indoor path network. This classification merely considers the edges and nodes that exist in the indoor path network and will contribute to 3D modeling of the path network. Hence, space concepts such as rooms will not be involved here. The edge and node types in an indoor path network are shown in Tables 1 and 2.

**Table 1.** Edge types in an indoor path network.

| ID | Edge Type |
|----|-----------|
| E1 | Corridor |
| E2 | Stairs |
| E3 | Elevator shaft |
| E4 | Corridor before the stairs |
| E5 | Corridor between the stairs |
| E6 | Corridor before the elevator |

**Table 2.** Node types in an indoor path network.

| ID | Node Type |
|----|-----------|
| N1 | Corridor junction |
| N2 | Corridor—corridor before the stairs |
| N3 | Corridor—corridor before the elevator |
| N4 | Corridor before the stairs—stairs |
| N5 | Corridor between the stairs—stairs |
| N6 | Corridor before the elevator—elevator |
| N7 | Elevator shaft—floor |

The test indoor paths used in this paper were drawn in a 3D indoor scene using an interactive tool that would output indoor paths as a shapefile (*.shp) file, which contained the paths' geographical coordinates (latitude, longitude, and elevation). Afterwards, paths and nodes were split apart according to their types defined above. After the *.shp file was imported into PostGIS, the software was used to create the topology of indoor paths. The topology construction process resulted in two tables where the geographic information of nodes and edges, such as id, geographic coordinates, source id, target id, etc., was stored. To apply path analysis algorithms and 3D path modeling algorithms, the attributes of nodes and edges (for edges, name, type, cost, and reverse cost, and for nodes, name and type) were set according to the actual situation and the classifications listed in Tables 1 and 2. This indoor path network contained 101 edges and 88 nodes, shown in Figure 1 as a traditional (3D lines and points) visualization.

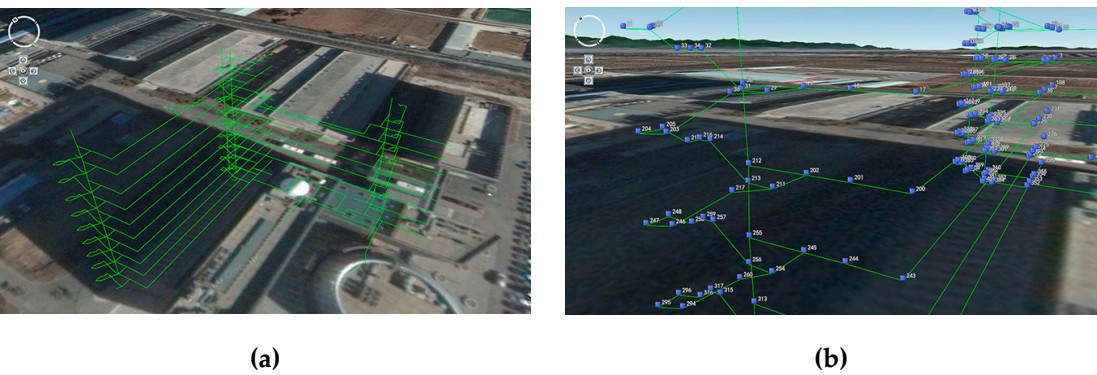

|  |  |
|:---:|:---:|
| (**a**) | (**b**) |

**Figure 1.** Indoor path network in a 3D virtual scene. (**a**) Overall view. (**b**) Edges and nodes.

### 3.2. Design of Basic Elements in Indoor Paths

The basic elements in indoor paths are divided into four categories: Corridor, stair, elevator and node.

(1) Corridor element. Indoor corridors are displayed as 3D polygons or cuboids. As shown in Figure 2, the blue solid line is a 3D corridor, and the black lines are edges of the 3D corridor model generated according to user-defined width (2W) and height (H).

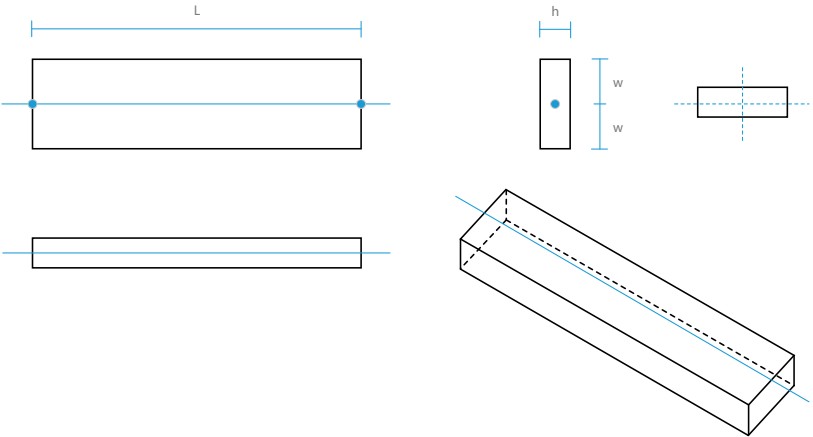

**Figure 2.** Design of a corridor element.

(2) Stair element. A stair element can be displayed simply as a 3D slope, or a solid model according to its original shape, as shown in Figure 3. The width of a stair element is the same as that of a corridor element, and the length of the slope is equal to the 3D distance between the two endpoints of the stair line segment.

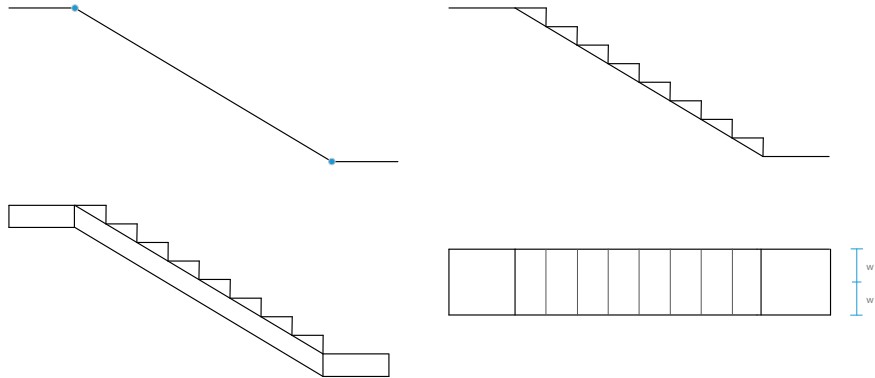

**Figure 3.** Design of a stair element.

(3) Elevator element. An elevator element, as shown in Figure 4, is displayed as a 3D box of size matching that of the elevator in the real world.

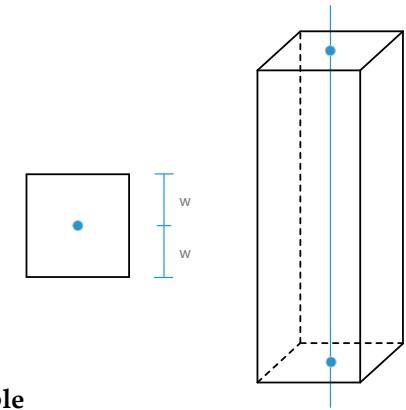

**table**

**Figure 4.** Design of an elevator element.

(4) Node element. An indoor path network may contain various kinds of nodes (N1 to N7, Table 2), but most of them (N2 to N7) can connect to other elements directly. Corridor junctions, such as intersection points and turning points, are abstracted into circular nodes and arc nodes. They can be displayed as cylinders and 3D sector models, as shown in Figure 5.

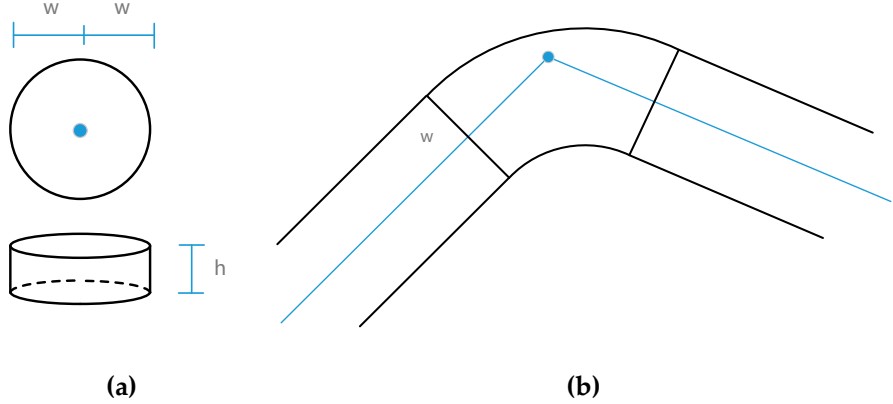

     **(a)**            **(b)**

**Figure 5.** Design of a node element. (**a**) Circular node. (**b**) Arc node.

### 3.3. Quaternion-based Calculation of Coordinates of Basic Elements

#### 3.3.1. Quaternion Operations

When calculating vertex coordinates with quaternions, many complicated and cumbersome trigonometric functions become unnecessary, and the angle interpolation calculation becomes more convenient. Although the quaternion rotation is in four dimensions, we do not need to know the specifics of internal calculations, and instead only set the four values of the angle and rotation vector. We can simply regard a quaternion as a rotation of $\theta$ degrees around a vector $(x, y, z)$ in three dimensions.

Denoting the rotation vector by $\vec{v}(x_v, y_v, z_v)$ and rotation angle by $\theta$, the calculation process of the corresponding quaternion $Q(x, y, z, w)$ and rotation matrix $R_Q$ is as follows:

$$\vec{n} = \vec{v} / \sqrt{x_v^2 + y_v^2 + z_v^2} \tag{3}$$

$$Q(x, y, z, w) = f\{\theta, \vec{v}\} = (\vec{v} \cdot \sin(\frac{\theta}{2}), \cos(\frac{\theta}{2})) \tag{4}$$

$$R_Q = \begin{bmatrix} 1 - 2(y^2 + z^2) & 2(xy - zw) & 2(xz + yw) & 0 \\ 2(xy + zw) & 1 - 2(x^2 + z^2) & 2(yz - xw) & 0 \\ 2(xz - yw) & 2(yz + xw) & 1 - 2(y^2 + y^2) & 0 \\ 0 & 0 & 0 & 1 \end{bmatrix} \tag{5}$$

Additionally, a quaternion can also be represented with a start vector $\vec{v_f}(x_f, y_f, z_f)$ and an end vector $\vec{v_t}(x_t, y_t, z_t)$, i.e., vector $\vec{v_f}$ becomes $\vec{v_t}$ after rotating by a certain angle. Then, quaternion $Q(x, y, z, w)$ is calculated as follows:

$$\vec{n_f} = \vec{v_f} / \sqrt{x_f^2 + y_f^2 + z_f^2} \tag{6}$$

$$\vec{n_t} = \vec{v_t} / \sqrt{x_t^2 + y_t^2 + z_t^2} \tag{7}$$

$$s = \sqrt{(1 + \vec{n_f} \cdot \vec{n_t})/2} \tag{8}$$

$$\vec{v_x} = \vec{v_f} \hat{\vec{v_t}} \tag{9}$$

$$Q(x, y, z, w) = f\{\vec{v_f}, \vec{v_t}\} = (\vec{v_x}/(2 \cdot s), s) \tag{10}$$

An important advantage of quaternion rotation is the ability to achieve smooth spherical interpolation. Assuming that two different quaternions $Q_f$ and $Q_t$ are evenly interpolated into $n$ parts, quaternion $Q_i$ $(i = 1, 2, \ldots, n)$ is calculated as follows:

$$\theta = arc\cos(Q_f \cdot Q_t) \tag{11}$$

$$s_f = \sin((1 - i/n) \cdot \theta)/\sin\theta \tag{12}$$

$$s_t = \sin(i/n \cdot \theta)/\sin\theta \tag{13}$$

$$Q_i = slerp(i/n, Q_f, Q_t) = Q_f \cdot s_f + Q_t \cdot s_t \tag{14}$$

#### 3.3.2. Calculation of Coordinates of a Line Segment Vertex

The first step of 3D modeling of an indoor path network is to calculate the vertex coordinates of each basic element of the paths, and then connect the vertices to form a 3D polygon or cuboid. If the coordinates were calculated by trigonometry, the angular relations would make the process cumbersome and difficult to deduce. As shown in Figure 6, $P_0(x_0, y_0, z)$ and $P_1(x_1, y_1, z)$ are two nodes

in the indoor paths; the length of $P_0P_1$ is L, and $P_1 - P_0 = (a, \vec{b}, 0)$; line AB is perpendicular to $P_0P_1$ and has the length of 2w. Coordinates of A and B are calculated by trigonometry as follows.

$$K_{AB} = -\frac{a}{b} \tag{15}$$

$$\sin \alpha = \frac{a}{L}, \cos \alpha = \frac{b}{L} \tag{16}$$

$$\Delta x = w \cdot \cos \alpha = w\frac{b}{L} \tag{17}$$

$$\Delta y = w \cdot \sin \alpha = w\frac{a}{L} \tag{18}$$

$$P_A = (x_0 - \Delta x, y_0 + \Delta y, z) \tag{19}$$

$$P_B = (x_0 + \Delta x, y_0 + \Delta y, z) \tag{20}$$

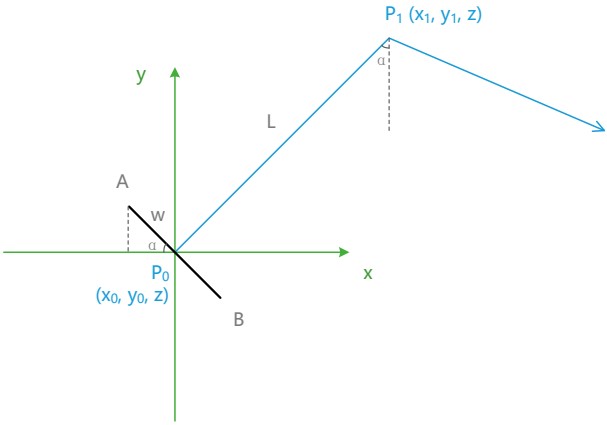

**Figure 6.** Example of calculation of coordinates by trigonometry.

If quaternions are used in the calculation, it will become easier to understand, and the process will be simpler. The basic idea of calculating coordinates with quaternions is to use the final coordinates as a result of rotation of a certain point or a line segment. As shown in Figure 7, line $A_0B_0$ is on the $x$ axis and has the same length as that of AB; then, AB can be considered a $\alpha$-degree clockwise rotation of $A_0B_0$ in the negative direction of the $z$ axis $(0, 0, -1)$.

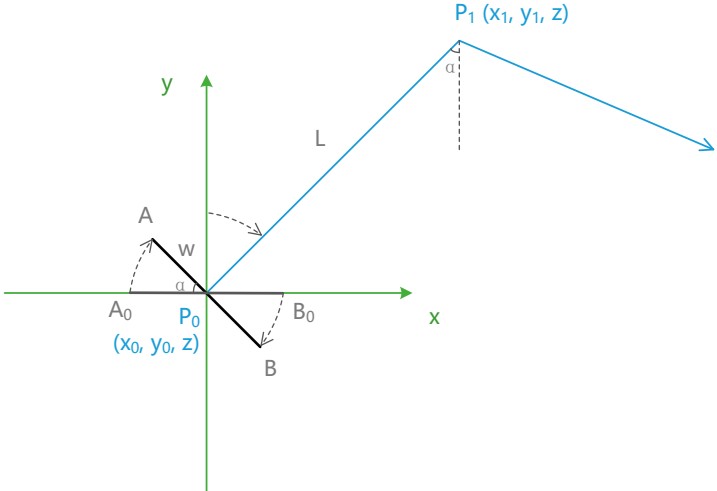

**Figure 7.** Example of calculating coordinates with quaternions.

Considering that axis $y$ ($\vec{y}$) is normal to $A_0B_0$, and $\overrightarrow{P_0P_1}$ is normal to $AB$, the rotation angle from $\vec{y}$ to $\overrightarrow{P_0P_1}$ is the same as that from $AB$ to $A_0B_0$, and so is the quaternion. Suppose that $Q_{AB}$ is the quaternion from $AB$ to $A_0B_0$, $M_{AB}$ is the rotation matrix of $AB$, and the coordinates of $A_0$ and $B_0$ are $(-w, 0, z)$ and $(w, 0, z)$; then, the coordinates of A and B are calculated as follows:

$$Q_{AB} = f\{\vec{y}, \overrightarrow{P_0P_1}\} = f\{(0,1,0), \overrightarrow{P_0P_1}\} \tag{21}$$

$$M_{AB} = Rotate(Q_{AB}) \tag{22}$$

$$P_A = (-w, 0, z) \cdot M_{AB} \tag{23}$$

$$P_B = (w, 0, z) \cdot M_{AB} \tag{24}$$

The above process shows that we need to neither care about the complex angular relationship between line segments nor perform any trigonometric operations when calculating the coordinates using quaternions.

If the middle point of the line segment is not at the origin, as shown in Figure 8, we can establish a local rectangular coordinate system at point $P_1$ and calculate the local coordinates of A and B in $x'P_1y'$; then, the coordinates can be translated according to the coordinates of $P_1$.

$$P_A = (-w, 0, 0) \cdot M_{AB} \cdot Trans(P_1) \tag{25}$$

$$P_B = (w, 0, 0) \cdot M_{AB} \cdot Trans(P_1) \tag{26}$$

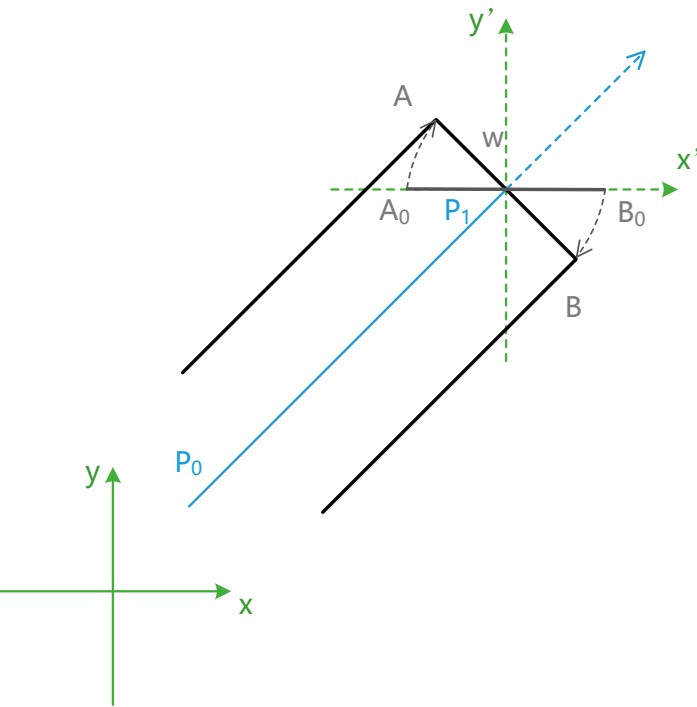

**Figure 8.** The case of the middle point of AB not being at the origin.

### 3.3.3. Calculation of Coordinates of Circular and Arc Nodes

Every point on a circular node can be considered a clockwise rotation of $PA_0$ (0, w, z) in the negative direction of the $z$ axis (0, 0, $-1$), as shown in Figure 9. If there are $n$ points on the circular node, the coordinates of the $i^{th}$ ($i$ = 1, 2, 3, ... , $n$) point ($P_A$) are then calculated as follows:

$$Q_A = f\{\alpha, -\overrightarrow{z}\} = f\{i \cdot 2\pi/n, (0, 0, -1)\} \tag{27}$$

$$P_A = (0, w, 0) \cdot Rotate(Q_A) \cdot Trans(P) \tag{28}$$

For an arc node (as shown in Figure 10, a corridor junction of $P_0P_1$ and $P_1P_2$), we should create two lines (AB and CD) before and after the intersection point $P_1$, and then generate the arc points by uniform interpolation from AB to CD.

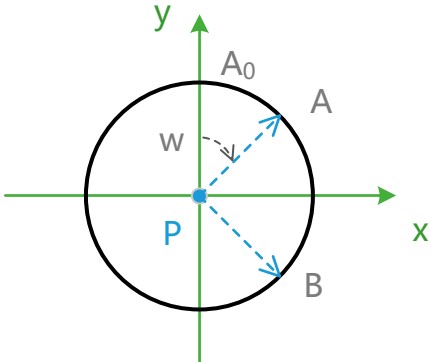

**Figure 9.** Calculation of coordinates of a circular node.

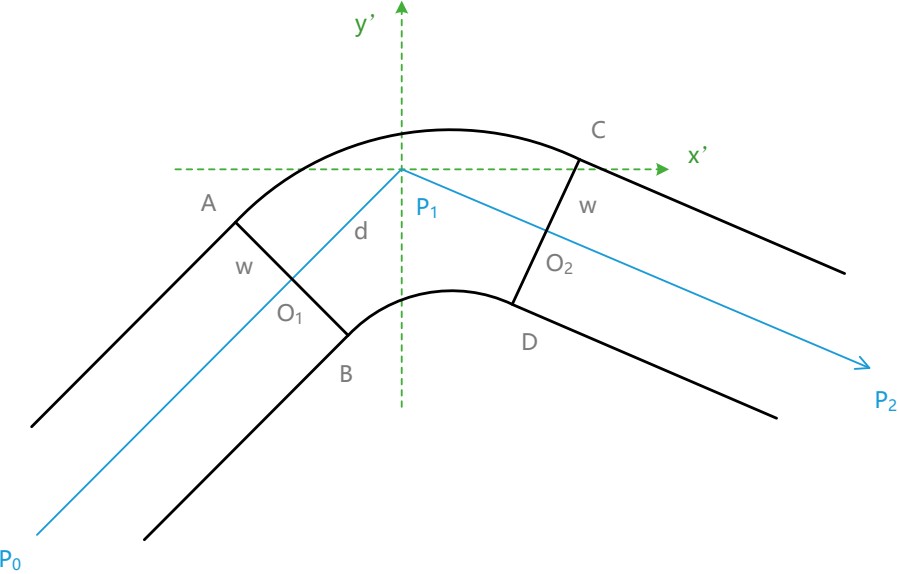

**Figure 10.** Calculation of coordinates of an arc node.

Suppose that $AB \perp P_0P_1$, $CD \perp P_1P_2$, $P_1O_1 = P_1O_2 = d$, and $x'P_1y'$ is a local rectangular coordinate system; then, AB can be considered an $\alpha$-degree clockwise rotation of $A_0B_0$, which is under the $x'$ axis and perpendicular to the $y'$ axis. The distance from $A_0B_0$ to $P_1$ is d, and AB = $A_0B_0$ = 2w. Similarly, CD can be considered a clockwise rotation of $C_0D_0$, which is above the $x'$ axis, as shown in Figure 11.

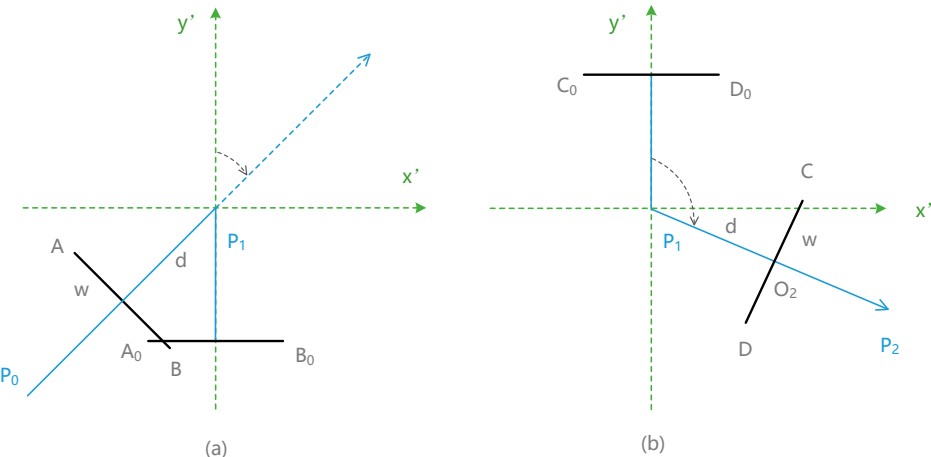

**Figure 11.** Quaternion rotations at an arc node.

Coordinates of $A_0$ and $B_0$ are $(-w, -d, z)$ and $(w, -d, z)$, and coordinates of A and B are calculated as follows:

$$Q_{AB} = f\{\vec{y}, \overrightarrow{P_0P_1}\} = f\{(0,1,0), \overrightarrow{P_0P_1}\} \tag{29}$$

$$P_A = (-w, -d, 0) \cdot Rotate(Q_{AB}) \cdot Trans(P_1) \tag{30}$$

$$P_B = (w, -d, 0) \cdot Rotate(Q_{AB}) \cdot Trans(P_1) \tag{31}$$

Similarly, coordinates of C and D are:

$$Q_{CD} = f\{\vec{y}, \overrightarrow{P_1P_2}\} = f\{(0,1,0), \overrightarrow{P_1P_2}\} \tag{32}$$

$$P_C = (-w, d, 0) \cdot Rotate(Q_{CD}) \cdot Trans(P_1) \tag{33}$$

$$P_D = (w, d, 0) \cdot Rotate(Q_{CD}) \cdot Trans(P_1) \tag{34}$$

Sector ABCD (with outer arc AC and inner arc BD) is composed of *n* interpolated lines, such as EF, all of which have equal length (2w) and go through the same center O, as shown in Figure 12. If the length of the sector radius ($OO_1$) is r, then the lengths of the inner and outer radii are r-w and r+w, respectively. As the rotation angle from $\overrightarrow{P_0P_1}$ to $\overrightarrow{P_1P_2}$ is the same as $\angle AOC$, the length of $P_1O_1$ (d) and the coordinates of O are calculated as follows:

$$Q_O = f\{\overrightarrow{P_0P_1}, \overrightarrow{P_1P_2}\} \tag{35}$$

$$\alpha = \angle AOE = Ang(Q_O)/2 \tag{36}$$

$$d = r \cdot \tan\alpha \tag{37}$$

$$P_O = (w + r, -d, 0) \cdot Rotate(Q_{AB}) \cdot Trans(P_1) \tag{38}$$

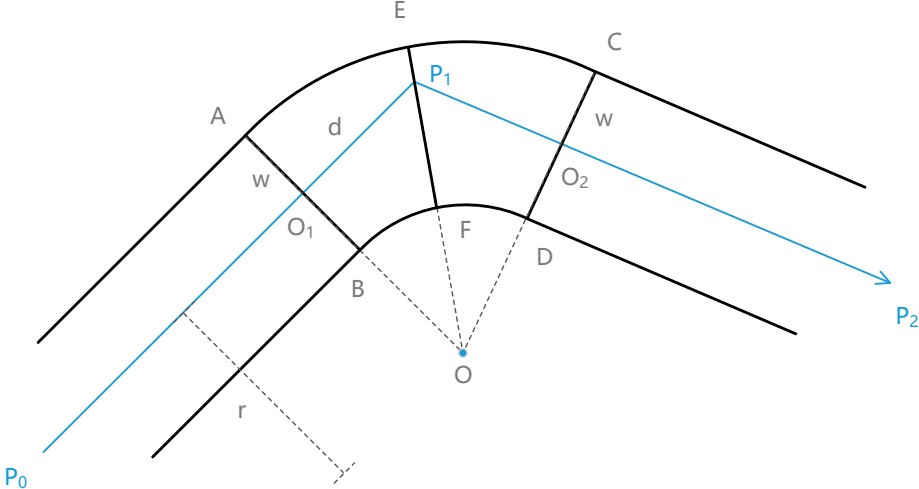

**Figure 12.** Calculation of coordinates by quaternion interpolation.

Sector ABCD is interpolated using quaternions according to the length of OB and OA; then, the quaternion and the coordinates of EF for the $i^{\text{th}}$ ($i = 1, 2, \ldots, n$) interpolated line are calculated as follows:

$$Q_{EF} = slerp(i/n, Q_{AB}, Q_{CD}) \tag{39}$$

$$P_E = (-r - w, 0, 0) \cdot Rotate(Q_{EF}) \cdot Trans(O) \tag{40}$$

$$P_F = (-r + w, 0, 0) \cdot Rotate(Q_{EF}) \cdot Trans(O) \tag{41}$$

### 3.3.4. Calculation of Coordinates of Stair Elements

Coordinates of stair elements can be calculated by quaternion rotation and translation by establishing a local rectangular coordinate system at a stair vertex. As shown in Figure 13, $P_0$ and $P_1$ are endpoints of a stair element. For each step in the stair, we establish a local rectangular coordinate system at the middle point of the step line; $P_0P_1'$ is the projection of $P_0P_1$ on the *xoy* plane. The coordinates of every step vertex are calculated by quaternion rotation from $P_0P_1'$ to the *y* axis.

$$\Delta step = (P_{P1} - P_{P0})/n \tag{42}$$

$$P_O = P_{P0} + i \cdot \Delta step \tag{43}$$

$$Q_O = f\{\vec{y}, \overrightarrow{P_0P_1'}\} = f\{(0, 1, 0), \overrightarrow{P_0P_1'}\} \tag{44}$$

$$P_A = (-w, 0, 0) \cdot Rotate(Q_O) \cdot Trans(P_O) \tag{45}$$

$$P_B = (w, 0, 0) \cdot Rotate(Q_O) \cdot Trans(P_O) \tag{46}$$

Similarly, coordinates of C and D are:

$$P_C = (-w, 0, 0) \cdot Rotate(Q_O) \cdot Trans(P_O + (0, 0, \Delta step.z)) \tag{47}$$

$$P_D = (w, 0, 0) \cdot Rotate(Q_O) \cdot Trans(P_O + (0, 0, \Delta step.z)) \tag{48}$$

If $P_0.z > P_1.z$, $P_0P_1$ points down, and the height of CD is the same as that of AB; then, coordinates of C and D are:

$$P_C = (-w, 0, 0) \cdot Rotate(Q_O) \cdot Trans(P_O + (\Delta step.x, \Delta step.y, 0)) \tag{49}$$

$$P_D = (w, 0, 0) \cdot Rotate(Q_O) \cdot Trans(P_O + (\Delta step.x, \Delta step.y, 0)) \tag{50}$$

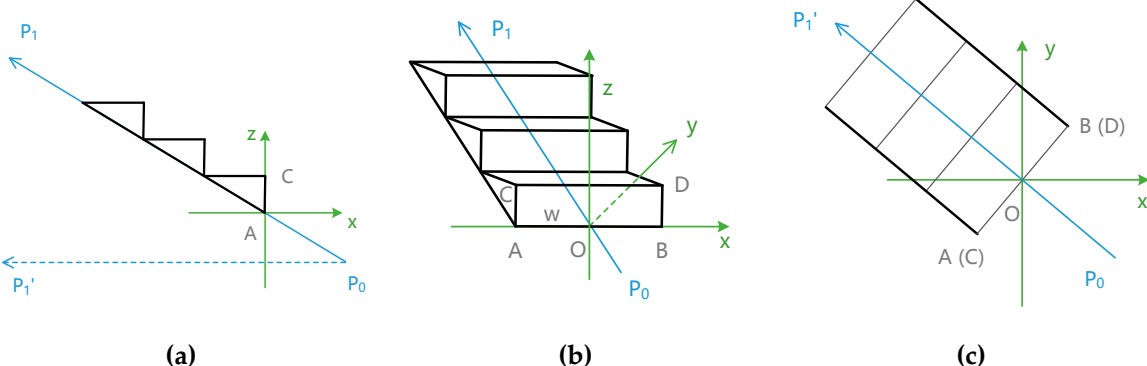

**Figure 13.** Calculation of coordinates of stair elements. (**a**) Side view. (**b**) Stereoscopic view. (**c**) Top view.

### 3.4. Piecewise 3D Modeling of an Indoor Path Network

Considering that the nodes and edges in the indoor path network have different shapes and ways of calculating coordinates, it is better to build the 3D indoor path model piecewise and successively, i.e., by processing nodes and edges in order, and building 3D models by the corresponding coordinate calculation and vertex organization methods. Some parameters, such as width, height, polygon type and model texture, can be customized by users or modelers. The steps of the piecewise method of 3D modeling of an indoor path network are shown in Figure 14.

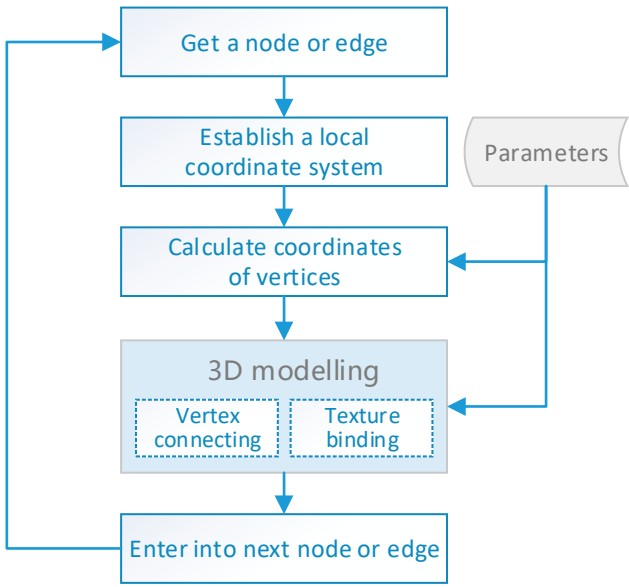

**Figure 14.** Piecewise 3D modeling method of an indoor path network.

The steps of this method can be stated as follows:

1.  Preparation. Implement the calculation of coordinates, connection of vertices, and texture binding algorithms for each basic element. The types of nodes and edges defined in Tables 1 and 2 should be considered because different types correspond to different implementations. Define the parameters of path models, including width, height, stair steps, circle sectors, polygon type, and model texture.
2.  Retrieve a node or edge from tables in the spatial database, read its attributes that will be used in the 3D modeling process, including id, geometric coordinates, type, source id and target id.

3. Establish a local coordinate system according to the geometric coordinates, and convert the coordinates from the geodetic (latitude, longitude, and height) coordinate system to the local Cartesian (ECEF X, Y, Z) coordinate system [29].
4. Calculate coordinates of vertices by the algorithms proposed above.
5. 3D modeling of paths and nodes. Connect the vertices together and bind the texture image to the model in the 3D GIS rendering engine.
6. Repeat steps 2-5 until all the edges and nodes have been processed.

## 4. Results

### 4.1. 3D Modeling of an Indoor Path Network

An example of piecewise 3D modeling of an indoor path network in a 3D Cartesian coordinate system is shown in Figure 15, and the values of the optional parameters used in the example are listed in Table 3.

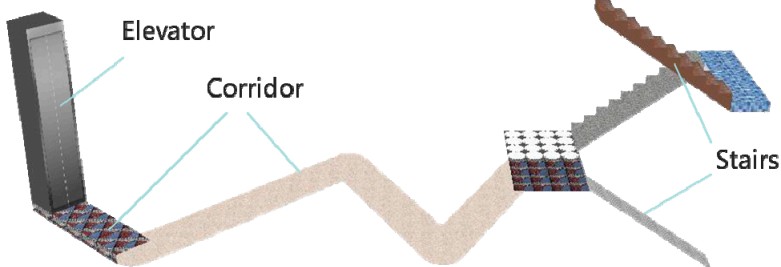

**Figure 15.** Piecewise 3D modeling of the basic elements in a 3D scene.

**Table 3.** Optional parameters and their values.

| Option | Value |
|---|---|
| Modeling Mode | Polygon or cuboid |
| Width | 2 meters |
| Height | 0.2 meters |
| Number of stair steps | 12 |
| Number of circle sectors | 30 |
| Textures | |

When generating 3D models from an indoor path network in a 3D GIS system, the earth curvature has to be considered because the 3D models will be inclined if they are created as a whole. Hence, all the coordinates of nodes and edges should first be converted from the geodetic coordinate system to the Cartesian coordinate system, transformed into 3D models separately, and then connected into an entire model of indoor paths. The result of piecewise 3D modeling of an indoor path network is shown in Figure 16. Compared with 3D lines shown in Figure 1, the 3D indoor path models can indicate the structure and types of indoor paths clearly with different shapes and textures.

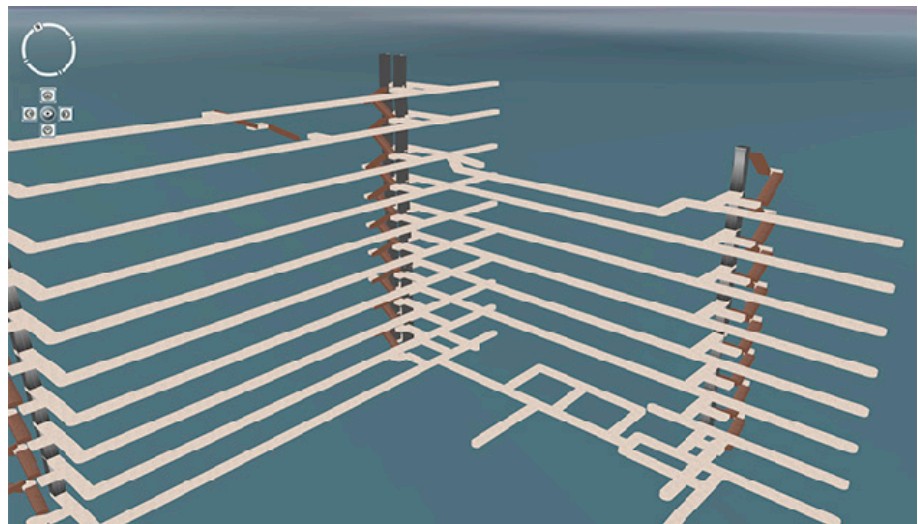

**Figure 16.** Piecewise 3D modeling of an indoor path network in a 3D virtual scene.

*4.2. Visualization of 3D Path Analysis with Indoor Path Models*

As 3D models of nodes and edges are generated separately, they can be linked together to form arbitrary routes, e.g., the shortest path. Thus, 3D indoor path models can also be used to visualize the results of 3D path analysis similarly to how it is done with 3D lines. Instead of highlighted lines, the shortest path will consist of a series of highlighted continuous 3D models.

In this paper, the 3D path analysis was performed in PostGIS by using pgRouting functions (pgr_dijkstra, pgr_astar, etc.) [30–32]. The result of path analysis is a table that contains the sequence ids of nodes and edges in the shortest path. Using these ids, the 3D models in the shortest path were retrieved from the indoor path models and then blended with a bright color using a fragment shader.

As shown in Figure 17, the shortest path calculated by the Dijkstra algorithm is highlighted in red in the indoor path models. The direction and type of each section are easily identifiable in the 3D scene.

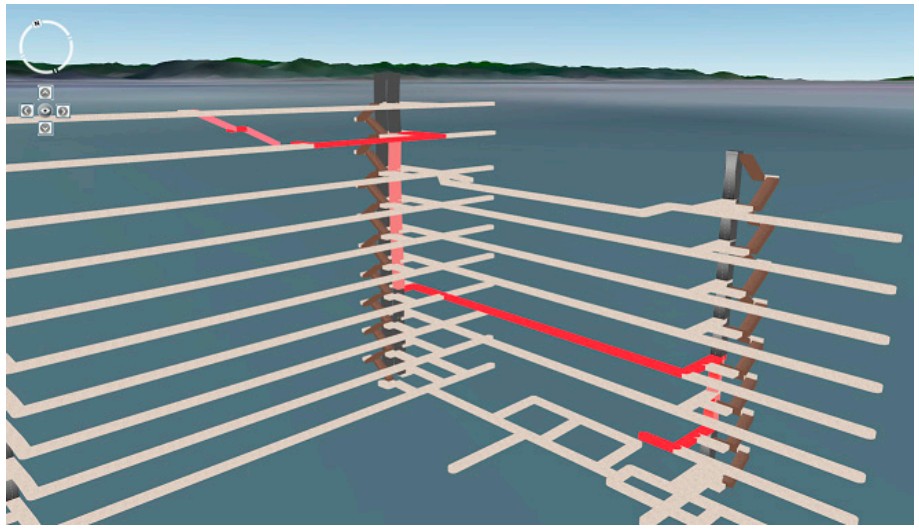

**Figure 17.** Visualization of a 3D path analysis with indoor path models.

*4.3. Efficiency Analysis*

The numbers of five 3D scene primitives (vertices, lines, polygons, triangles and quadrangles) in three different visualization modes (no paths, paths visualized as lines and paths visualized as

models), which can indicate the differences of newly generated entities and can be used as a reference for comparing the frame rates, were recorded. Before loading the indoor paths, the 3D scene mainly contains the meshes of the earth surface. The scene with 3D lines shown in Figure 1a contains 984 vertices and 492 lines but no triangles or quadrangles, as listed in Table 4. The scene with 3D models of indoor paths contains 34,710 vertices, 9,672 polygons and 1,512 triangles but no lines; these counts are not particularly large compared with those of triangular mesh-based models [33]. This observation means that the proposed quaternion-based 3D modeling method can generate detailed and irredundant models of indoor path networks.

**Table 4.** Numerical comparison of 3D scene primitives in various visualization modes.

| Mode | Vertices | Lines | Polygons | Triangles | Quadrangles |
|------|----------|-------|----------|-----------|-------------|
| None | 34252 | 0 | 0 | 47168 | 1 |
| Lines | 35236 | 492 | 0 | 47168 | 1 |
| Models | 68962 | 0 | 9672 | 48680 | 127 |

The frame rates of four stages (no paths, paths visualized as lines, paths visualized as models and path analysis with 3D models) were recorded consecutively when the 3D GIS was running on a laptop with an NVIDIA GeForce GT 730M graphics card with 1GB of onboard RAM. As shown in Figure 18, the blue solid line represents the real-time frame rate, and the orange dotted line represents the change trend of the frame rate. Before the indoor paths were loaded, the system ran a 3D scene containing 34,000 vertices and 47,000 triangles at approximately 60 fps, and the visualization of 3D lines did not affect the frame rates (Stages 1 and 2, 1 to 21s). The frame rate dropped to 39 fps when 3D models were being generated but soon rose to 54 fps after the process was completed (Stage 3, 22 to 37s). The 3D path analysis algorithm did not affect the frame rate, and the efficiency of rendering highlighted paths was the same as that of rendering 3D models in Stage 3. Although the frame rate dropped by approximately 3 fps to 10 fps when the path network was being represented as 3D models, the frame rate could still remain above 50 fps in a 3D scene containing 68,000 vertices, 9000 polygons and 48,000 triangles.

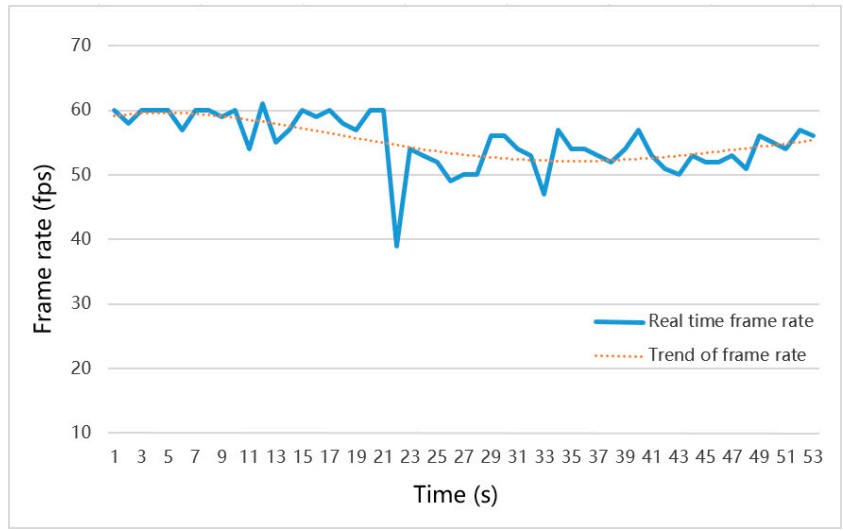

**Figure 18.** Frame rates of 3D path analysis with indoor path models. Stage 1: no lines, 1–10s; Stage 2: paths visualized as lines, 11–21s; Stage 3: paths visualized as models, 22–37s; Stage 4: path analysis with 3D models: 38–53s.

## 5. Discussion

This paper proposed a quaternion-based piecewise 3D modeling method to automatically generate 3D models of indoor paths to support and improve the visualization performance of 3D indoor path analysis. To create 3D models of indoor path networks, the paths were classified into four types of basic elements: Corridor, stair, elevator and node, which contained six kinds of edges and seven kinds of nodes. A quaternion-based method was proposed to calculate the coordinates of the designed elements, and a piecewise 3D modeling method was implemented to create the entire 3D indoor path models in the 3D GIS scene. The numbers of 3D scene primitives indicated that the proposed quaternion-based 3D modeling method could generate detailed and irredundant models of indoor path networks. The result of 3D path analysis showed that indoor path models could not only provide a better visualization performance than that of 3D lines by displaying a path with different shapes, textures and colors, but also maintain a high rendering efficiency (above 50 fps) in a 3D scene containing more than 50,000 polygons and triangles. Thus, the proposed method will be very useful for visualizations of indoor path networks in 3D GIS, especially when there are no detailed 3D building models or when the types and structures of 3D indoor paths are viewed separately.

Although this method can generate 3D models from indoor paths automatically and highlight the chosen/optimal paths vividly, the indoor path network should be adequately prepared, for instance, by splitting the paths according to their type and by perfecting the attributes (type, cost used in path analysis, etc.) of edges and nodes in the spatial database. An interactive tool should be created to assist with these tasks in future. In addition, the texture coordinates and normal directions of vertices should be handled carefully; otherwise, the textures of models will be distorted or invalid.

**Author Contributions:** Conceptualization, H.J.; data curation, X.K.; formal analysis, J.L. and Q.J.; funding acquisition, X.F.; methodology, H.J.; project administration, X.F.; software, H.J.; supervision, X.F.; writing—original draft, H.J. and J.L.; writing—review & editing, X.F.

**Funding:** This research was funded by the Strategic Priority Research Program of the Chinese Academy of Sciences, grant No. XDA19080101, the National Key Research and Development Program of China, grant Nos. 2016YFB0501503 and 2016YFB0501502, and Hainan Provincial Department of Science and Technology, grant No. ZDKJ2016021.

**Acknowledgments:** The authors would like to thank the editors and anonymous reviewers for their valuable comments and suggestions. We would also like to thank Xiaoping Du, Zhenzhen Yan, Qin Zhan and Junjie Zhu for helpful discussions.

**Conflicts of Interest:** The authors declare no conflicts of interest. The funders had no role in the design of the study, in the collection, analyses, or interpretation of data, in the writing of the manuscript, or in the decision to publish the results.

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
