# Peer review of "A Quaternion-Based Piecewise 3D Modeling Method for Indoor Path Networks"

_ijgi, doi:10.3390/ijgi8020089_

Round 1
Reviewer 1 Report
After Section "Introduction" I would have expected a Section "Related Work" or "Background"
Section 2, consisting of one paragraph, should be a subsection of something like "Concept".
Figure 1 is not referenced in the text?
The Figures are briliant and the explanation is very fine, very didactive. However, the structure of the paper is a bit strange. Like I said, I expect a background section, related work. Then your concept and afterwards your evaluation
Section 3 would also be a subsection...
The biggest weak point is, that there is no evaluation. You show the very nice result in figure 16 and then in Figure 17 a highlighted dijkstra path. And then you state that this is "a better visualization performance" (Section 6, First Paragraph, last sentence).
Why do you have exactly 4 node types? Why not hall? Crossing? Room?
The defined edges from section 2, did you use them after you defined them?
Author Response
Response to Reviewer 1 Comments
Point 1: The structure of the paper: After Section "Introduction" I would have expected a Section "Related Work" or "Background"; Section 2, consisting of one paragraph, should be a subsection of something like "Concept"; The Figures are briliant and the explanation is very fine, very didactive. However, the structure of the paper is a bit strange. Like I said, I expect a background section, related work. Then your concept and afterwards your evaluation; Section 3 would also be a subsection.
Response 1: It’s our fault not to organize the structure of the paper reasonably. In this revised manuscript we adjust the structure as “Introduction - Related Works - Materials, Concepts and Methods – Results - Discussion”. In Section 2 we supplemented the research background and related works including visualization of 3D indoor path analysis and 3D model generation methods from lines (Page 2 to 3). The concepts and methods proposed in this work are all described as subsections in Section 3.
Point 2: Figure 1 is not referenced in the text?
Response 2: Figure 1 demonstrated the materials of our work but indeed we did not reference it in the later text. In this manuscript we referenced in the results (Page 15, line 301; Page 16, line 327) to compare the 3D path models and 3D lines.
Point 3: The biggest weak point is, that there is no evaluation. You show the very nice result in figure 16 and then in Figure 17 a highlighted dijkstra path. And then you state that this is "a better visualization performance".
Response 3: In this manuscript we supplemented a subsection “4.3. Efficiency Analysis” (Page 16) in the results to evaluate our work. The numbers of five 3D scene primitives (Vertices, Lines, Polygons, Triangles and Quadrangles) in three different visualization modes (No paths, paths visualized as lines and paths visualized as models) were compared, which indicated that the proposed method can generate detailed and irredundant models for indoor path network. The frame rates of four stages (No paths, paths visualized as lines, paths visualized as models and path analysis with 3D models) were recorded consecutively in 3D GIS. The result showed that the method can keep a high rendering efficiency (above 50 fps) in a 3D scene containing more than 50 thousand polygons and triangles on a laptop which has an NVIDIA GeForce GT 730M graphics card with 1GB of memory.
Point 4: Why do you have exactly 4 node types? Why not hall? Crossing? Room?
Response 4: In this paper the indoor path network was classified into four types of basic elements: corridor, stair, elevator and node, which contain six kinds of edges and seven kinds of nodes. The nodes and edges were classified according to their shapes and functions in order to create detailed and vivid 3D models for the path network. This classification merely considers the edges and nodes which exist in the indoor path network and will contribute to 3D modelling for the path network. So the Space concepts such as Rooms will not be involved here. However, the concepts like Halls and Rooms are very useful in path navigation and guidance. They should be added to the spatial database as attributes of nodes and edges in order to create instructive paths for navigation and guidance.
Point 5: The defined edges from section 2, did you use them after you defined them?
Response 5: The classification of edges and nodes plays an important role in this work and we feel so sorry not to describe how to use them. They will be used throughout the data preparation and processing process, including paths splitting and attributes improvement (Page 4, line 132 to 142). The types of nodes and edges should also be carefully considered in the piecewise 3D modelling procedure because different types correspond to different implementations (Page 14, line 273).

Reviewer 2 Report
This paper presents a a quaternion-based piecewise 3D modelling method was proposed to automatically generate highly recognizable 3D models for indoor path network. Overall this work is interesting. However, there are several issues which must be addressed. First, the work was not well motivated. The introduction is too short. I suggest the authors to think hard about why this work is needed. A good way is to put it in the context of smart cities. Some good references are: Houbing Song, Ravi Srinivasan, Tamim Sookoor, Sabina Jeschke, Smart Cities: Foundations, Principles and Applications. ISBN: 978-1-119-22639-0, Hoboken, NJ: Wiley, 2017, pp.1-906; Z. Lv, et al, "Virtual Reality Smart City Based on WebVRGIS," in IEEE Internet of Things Journal, vol. 3, no. 6, pp. 1015-1024, Dec. 2016. doi: 10.1109/JIOT.2016.2546307; Y. Sun, et al, "Internet of Things and Big Data Analytics for Smart and Connected Communities," in IEEE Access, vol. 4, pp. 766-773, 2016. doi: 10.1109/ACCESS.2016.2529723. Second, the applicability of the proposed work must be discussed properly.
Author Response
Point 1: The work was not well motivated. The introduction is too short. I suggest the authors to think hard about why this work is needed. A good way is to put it in the context of smart cities. Some good references are: Houbing Song, Ravi Srinivasan, Tamim Sookoor, Sabina Jeschke, Smart Cities: Foundations, Principles and Applications. ISBN: 978-1-119-22639-0, Hoboken, NJ: Wiley, 2017, pp.1-906; Z. Lv, et al, "Virtual Reality Smart City Based on WebVRGIS," in IEEE Internet of Things Journal, vol. 3, no. 6, pp. 1015-1024, Dec. 2016. doi: 10.1109/JIOT.2016.2546307; Y. Sun, et al, "Internet of Things and Big Data Analytics for Smart and Connected Communities," in IEEE Access, vol. 4, pp. 766-773, 2016. doi: 10.1109/ACCESS.2016.2529723.
Response 1: It is an excellent advice to apply our work to Smart City and the recommended references are very helpful for us to find the motivations and application significances of this work (Page 1, line 42; Page 2, line 66; Page 17, line 365-367). In this revised manuscript we expanded the Introduction and add a section “Related Works” to support our work. In the Introduction we described the background, motivations, contents and application prospects of this work. Section 2 introduces some related works including the visualization of 3D indoor path analysis and 3D model generation methods from lines.
Point 2: The applicability of the proposed work must be discussed properly.
Response 2: In this manuscript the applicability of this work was described in the Introduction (Page 2, line 64-66) and the Discussion (Page 17, line 365-373). We believe that our work will have promising prospects for path analysis related applications in smart cities and will be very useful for visualizations of indoor path network in 3D GIS especially when there are no detailed 3D building models or viewing the types and structures of 3D indoor paths separately. In addition, the application scenarios and limits of the proposed method was discussed at the end of the manuscript.
Point 3: English language and style: Moderate English changes required
Response 3: Indeed, the language and style of this manuscript needs a moderate revision. We are planning to ask MDPI or native English-speakers for help to make sure the article is fluent and easy to understand for readers.

Reviewer 3 Report
1. Very short literature review and list of references. The state-of-the-art and literature review are not sufficient.
2. The use of quaternions for animation, modelling, rendering and game development is well-known things. What’s new authors have contributed to this region by this paper?
3. In the paper authors proposed a method of interior space modelling in a building. In terms of terminology, it can be a bit confused with well-known navigation field of path planning. Maybe, it is better to substitute the words of “indoor path” in the title and the abstract?
4. No details about how the path was calculated by Dijkstra algorithm in the case for Figure 17.
5. No numerical evaluation of proposed method and comparison with other methods. What’s the advantage of proposed one?
6. The methodology in section 5 should be more fully described.
Conclusion: The paper contribution should be clarified after detailed literature review investigation. Introduction and methodology should be extended with more detailed explanation. The sections on numerical evaluation of proposed method and comparison with other methods must be added.
Author Response
Point 1: Very short literature review and list of references. The state-of-the-art and literature review are not sufficient.
Response 1: Indeed, the literature review and list of references of last manuscript are very short. In this revised manuscript we expanded the Introduction and add a section “Related Works” to support our work. In the Introduction we described the background, motivations, contents and application prospects of this work. Section 2 introduces some related works including the visualization of 3D indoor path analysis and 3D model generation methods from lines (Page 2 to 3).
Point 2: The use of quaternions for animation, modelling, rendering and game development is well-known things. What’s new authors have contributed to this region by this paper?
Response 2: It is true that quaternions have been widely used in modelling, rendering, animation and object control (Page 3, line 113). We are afraid this work will contribute little to the development of quaternions. However, we took the advantages of quaternion operation to automatically create 3D path models from indoor path networks in order to support and improve the visualization performance of 3D indoor path analysis.
Point 3: In the paper authors proposed a method of interior space modelling in a building. In terms of terminology, it can be a bit confused with well-known navigation field of path planning. Maybe, it is better to substitute the words of “indoor path” in the title and the abstract?
Response 3: We are appreciated for your concern about the possible confusion with navigation field of path planning. In this work we took the indoor path network (including paths and nodes) as an input of the proposed method to generate 3D indoor path models from the topological network, so as to support 3D path analysis and improve the visualization performance of the results as well. Thus we still used the same phrase in this manuscript.
Point 4: No details about how the path was calculated by Dijkstra algorithm in the case for Figure 17.
Response 4: It’s our fault not to describe the details about how the path was calculated by Dijkstra algorithm. In this revised manuscript we explained it in section “4.2. Visualization of 3D Path Analysis with Indoor Path Models” (Page 15, line 311-315). The 3D path analysis was done in PostGIS by calling pgRouting functions (pgr_dijkstra) The result of path analysis is a table that contains the sequence ids of nodes and edges in the shortest path. According to these ids the 3D models in the shortest path were retrieved from the indoor path models and then blend with bright red colour through a fragment shader.
Point 5: No numerical evaluation of proposed method and comparison with other methods. What’s the advantage of proposed one?
Response 5: In this manuscript we supplemented a subsection “4.3. Efficiency Analysis” (Page 15) in the results to evaluate our work. The numbers of five 3D scene primitives (Vertices, Lines, Polygons, Triangles and Quadrangles) in three different visualization modes (No paths, paths visualized as lines and paths visualized as models) were compared, which indicated that the proposed method can generate detailed and irredundant models for indoor path network. The frame rates of four stages (No paths, paths visualized as lines, paths visualized as models and path analysis with 3D models) were recorded consecutively in 3D GIS. The result showed that the method can keep a high rendering efficiency (above 50 fps) in a 3D scene containing more than 50 thousand polygons and triangles on a laptop which has an NVIDIA GeForce GT 730M graphics card with 1GB of memory.
Point 6: The methodology in section 5 should be more fully described.
Response 6: In this manuscript we supplemented the process of the piecewise 3D modelling procedure in section 3.4 (Page 14, Line 271-285) so that reader can easily understand how the method works.

Round 2
Reviewer 2 Report
The reviewers' comments have been addressed properly.
Author Response
Point 1: English language and style are fine/minor spell check required Response 1: We have summit the manuscript to American Journal Experts (AJE) for English editing services. Please review this revised version for detail.Reviewer 3 Report
I wonder why the Int. Journal of Geo-Science was chosen for this publication. Since the target audience of this journal is interested in remote sensing and photogrammetry, the paper will not attract most of readers and will not distribute the ideas to relevant audience.
A comparison with other methods could be a plus for this paper.
The proof of concept was provided by one 3D model case. The models and experiments should be more to demonstrate the method's robustness and workability.
Nevertheless, thanks for the paper improvements.
Author Response
Point 1: I wonder why the Int. Journal of Geo-Science was chosen for this publication. Since the target audience of this journal is interested in remote sensing and photogrammetry, the paper will not attract most of readers and will not distribute the ideas to relevant audience.
Response 1: The proposed method in this paper will support and improve the visualization performance of indoor path analysis in 3D GIS and meet the demand of path analysis related applications in Smart Cities. The reason why the Int. Journal of Geo-Information (not the Int. Journal of Geo-Science) was chosen for this publication was that the contents in this study fell inside the scope of this journal, e.g., “visualization theory and technology in real and virtual environments”, “spatial analysis, data mining, and decision support systems”, “applications of geoinformation technology (all possible domains)”, etc.. In addition, some related studies have published in this journal, as we cited in this paper, “Alattas, A.; Zlatanova, S.; Van Oosterom, P.; Chatzinikolaou, E.; Lemmen, C.; Li, K.J. Supporting Indoor Navigation Using Access Rights to Spaces Based on Combined Use of IndoorGML and LADM Models. Isprs Int J Geo-Inf 2017, 6, 384”, “Schabus, S.; Scholz, J.; Lampoltshammer, T.J. Mapping Parallels between Outdoor Urban Environments and Indoor Manufacturing Environments. Isprs Int J Geo-Inf 2017, 6, 281.”. Thus, we think this paper will attract readers who are interested in indoor path navigation and path analysis related applications in Smart Cities.
Point 2: A comparison with other methods could be a plus for this paper.
Response 2: We agree that comparison with different methods should be included in an article. The visualization of 3D indoor path analysis in the existing studies is mainly concerned with the (semantic) division and representation of indoor space, and the paths are usually visualized as 3D lines that cannot clearly indicate the types of the paths (Page 2, line 70 to 92). In this paper, the semantic idea was introduced to classify indoor path networks and the paths were visualized as 3D models (with textures). When generating 3D models from lines, a comparison of 3D modeling methods should be made in two respects: computational process and results of 3D modelling. Compared with Euler angle methods, quaternions are much more flexible and elegant in coordinate transformation and 3D modeling because users need to neither care about the complex angular relationship between line segments nor perform any trigonometric operations when calculating the coordinates using quaternions (Page 3, line 95 to 110; page 7 to 9, line 182 to 203). The results of proposed method, i.e., the indoor path models, are proved to be detailed enough and irredundant, and the numbers of five 3D scene primitives (vertices, lines, polygons, triangles and quadrangles) are not particularly large compared with those of triangular mesh-based models (Page 14 to 15, line 294 to 329).
Point 3: The proof of concept was provided by one 3D model case. The models and experiments should be more to demonstrate the method's robustness and workability.
Response 3: Indeed, the method has been tested in one 3D model case, but the indoor path network used in this paper has included all the edge types (E1 – E6) listed in Table 1 (Page 3, line 124). The statistics of edge types from the edge table in PostgreSQL is shown in the attachment. The method will generate 3D models from indoor paths automatically and highlight the chosen/optimal paths vividly as long as the input data (indoor path network) was processed properly as designed. However, as we discussed is Section 5 (Page 16, line 364 to 370), the indoor path network has to be adequately prepared, for instance, by splitting the paths according to their type and by perfecting the attributes (type, cost used in path analysis, etc.) of edges and nodes in the spatial database. An interactive tool should be created to assist with these tasks and more experiments should be done in future.
